# A Pest to Mental Health? Exploring the Link between Exposure to Agrichemicals in Farmers and Mental Health

**DOI:** 10.3390/ijerph16081327

**Published:** 2019-04-12

**Authors:** Nufail Khan, Alison Kennedy, Jacqueline Cotton, Susan Brumby

**Affiliations:** 1School of Medicine, Deakin University, 75 Pigdons Road, Waurn Ponds, VIC 3216, Australia; nufi_k@hotmail.com (N.K.); Alison.Kennedy@wdhs.net (A.K.); Jacquie.Cotton@wdhs.net (J.C.); 2National Centre for Farmer Health, Western District Health Service, Hamilton, VIC 3300, Australia

**Keywords:** agrichemical, pesticide, organophosphate, mental health, farming, suicide, low-level exposure, chronic toxicity, Total Worker Health

## Abstract

The current literature acknowledges that occupational exposures can adversely affect mental health. This review seeks to elucidate the current understanding of the effect of agrichemical exposure on mental health in the agricultural sector, including low-dose, chronic pesticide exposure. This scoping review adopted a snowballing and saturation approach. The review highlights inconsistencies in linking poor mental health and pesticide use. While some studies specifically showed that both high- and low-dose pesticide exposure were associated with poor mental health, consistent and rigorous research methods are lacking. The review also proposes terms to delineate exposure types described in the literature. The review outcomes direct efforts to protect the health, wellbeing and safety of farming communities across the globe.

## 1. Introduction

### 1.1. The Mental Health Burden

Poor mental health interferes with an individual’s emotional, social and intellectual capabilities [1,2]. Poor mental health is often driven by mental health disorders, which are a growing concern on the global scale, presenting a 37.6% increase in prevalence in merely two decades (1990–2010) [3,4]. Mental health disorders represent an umbrella term to encompass the various conditions that can impinge on poor mental health. This includes the two main diagnostic categories of mood disorders (such as depression) and anxiety disorders, which are amongst the most highly prevalent mental health disorders globally [5]. Of note, mental health disorders can be transient (state of mood), involve a clinically significant symptom or syndrome that may be identified by scoring tools and, of course, can be a diagnosed disorder under the Diagnostic and Statistical Manual of Mental Disorders —DSM-5) [6]. Overall, mental health disorders are a leading cause of morbidity globally, with a recent World Health Organisation (WHO) mental health survey from 2009 estimating that the lifetime risk of developing a mental health disorder is at least 30% [4].

Australian research recapitulates global findings, with the *2007 National Survey of Mental Health and Wellbeing* describing mental health disorders as contributing to ~12.9% of Australia’s burden of disease, which places it as the third largest contributor behind cardiovascular disease and cancer [7]. This burden is exemplified by the fact that 45% of Australians aged 16–85 years will be expected to meet the criteria for diagnosis of a mental health disorder at some stage in their life, with 20% of Australians meeting these criteria in any one year [8].

Evidence now exists to show there are groups of the population, such as particular occupations, that exhibit an increased incidence and greater impact of poor mental health. There is a complex mix of biological, psychological and social factors postulated to account for this [9,10], including chronic disease, socioeconomic factors and, in particular, occupational health and safety. With the latter in mind, studies have identified increased rates of mental health disorders to be associated with particular professions. A large study of 18,572 participants conducted in the U.S showed that farmers displayed the greatest level of major depression (as measured by the Diagnostic Interview Schedule—a structured interview used by non-health professionals using DSM-III diagnostic standards) when compared to all other occupations [11]. A subsequent and similarly large health study of 17,295 participants in Norway also found that farming, fishing and forestry as an occupation had the greatest incidence of anxiety- and depression-related symptoms (as measured by the Hospital Anxiety and Depression Scale) [11,12]. Additionally, a recent literature review by Klingelschmidt and colleagues (2018) found higher risk of suicide to exist in workers within the forestry, fishing and agricultural sectors compared to other work sectors [13]. Overall, the farming sector unfortunately appears to be associated with poor mental health, diagnosed mental health disorders and increased suicide risk—all of which are considered in this review.

### 1.2. Farmers’ Mental Health

Agriculture as an occupation carries inherent physical and psychological challenges. While a body of work alludes to heightened rates of poor mental health and diagnosed mental health disorders among farmers when compared to other occupations [14,15,16], other research reports lower rates of diagnosed depression compared with the non-farming population [14,17]. These mixed results may reflect the varied use of diagnostic tools [16] and small sample sizes [17].

Despite contentious literature on the rates of diagnosed mental health disorders, Australian and international data presents strong evidence that suicide rates are higher in farmers compared to other occupations [16,18,19,20,21], although these rates vary regionally [22,23]. Australian agricultural workers have been identified as having the highest suicide rates [18], which are twice the suicide rate for any other employed worker [18,23].

Recent research may help to shed further light on the mixed findings regarding the prevalence of diagnosed mental health disorders between farmers and non-farmers [24] in the presence of a concomitant greater risk of suicide [13,14,15,19,25]. Australian researchers (2017) recently attempted to contextualise the range of pathways that lead to male farmer suicide [15]. The analysis identified both *situational* and *protracted* pathways. A *situational* pathway to suicide was characterised by an acute period of interpersonal, financial or work-related stress without the necessary presence of a diagnosed mental health disorder. A *protracted* pathway was characterised by a longstanding and established mental health disorder. Such conclusions suggest that a diagnosed mental health disorder is not necessarily a precursor to suicide risk among farmers. Instead, the suicide risk in farmers may be more clearly understood by improving the understanding of the unique ‘situational’ risk factors and occupational exposures that agricultural work entails, including exposure to agrichemicals.

### 1.3. The Influence of Farming Life and Culture on Mental Health

Agriculture has been a major part of Australia’s identity since European colonisation [26], with approximately 60% of Australia’s land used for farming, supplying more than 90% of Australia’s required domestic produce [27,28]. Farming has been historically more physically demanding, involving longer work hours, with a reduced likelihood to seek respite or vacations off farm, and is more socially and spatially isolating than other occupations [29,30]. Farmers are frequently heralded as stoic and independent individuals and less likely than metropolitan populations to access support for health and wellbeing [24]. Despite the historical stoic view of a farmer, these identified factors may in fact predispose farmers—as an occupation—to being one of the most vulnerable populations with regard to a variety of mental health challenges.

It is now recognised that aspects of the farming/agricultural work environment such as financial instability, isolation, or exposure to chemicals can impinge on mental health [14,17,31]. By accepting that a worker’s environment can impact on mental health, the canonical public health measures towards occupational health and safety now require a holistic view to capture the non-canonical causes of poor health and safety in the farming workplace.

The Total Worker Health (TWH)™ approach in the US has garnered significant interest due to the ability to consider the work environment as a potential health hazard [32,33,34,35,36]. Of great interest here is the strong understanding and integration of psychological factors in this interventional program [37]. Thereby, the TWH™ goes beyond the traditional health and safety paradigms by promoting mental health wellbeing. Overall, this is similar to the Sustainable Farm Families™ approach used in Australia and, more recently, in Canada, where farmers firstly consider themselves, then their family and then the farm business through a health, wellbeing and safety lens [38,39,40]. Given the complexity of contributing factors to poor mental health in farmers, a holistic and TWH approach behoves us to include consideration of the impact of the environment in which farmers live and work on their health and wellbeing. This includes exposure to agrichemicals—the focus of this review.

### 1.4. Agrichemical Exposure in Farming Populations

As mentioned afore, the farming culture and its practices can impact mental health; however, a particular area of concern herein is the exposure of agricultural workers to farming chemicals (or agrichemicals) such as pesticides. Agrichemicals target a number of pests that may threaten the yield of agricultural products such as broad acre crops, horticulture and animal products [41,42]. Conversely, improper use may predispose farmers to the negative effects of undesired exposure [43]. Organophosphate-based pesticides not only are one of the most widely used group of insecticides but also are considered by the WHO as one of the most harmful to humans, particularly where unsafe practices occur, e.g., in developing countries [44,45].

The potential for exposure to agrichemicals is without doubt an omnipresent issue. With this in mind, it is worrisome that the farming culture around agrichemical use lends to hazardous practices. This may be exacerbated, in developing nations, by poor knowledge about pesticides [46,47], low literacy levels reducing the use of personal protective equipment [48,49], inadequate storage and inappropriate disposal of pesticides [50,51], all of which make farmers susceptible to agrichemical exposure. Australian data also suggests that adherence to historical unsafe farming practices, particularly on multi-generational farms, increases the exposure risk to agrichemicals [52], as does the acceptance of high-risk behaviours as the norm on farms [52].

Pesticide exposure can influence many health outcomes. The focus of this review is to explore the potential link between pesticide exposure and development of poor mental health in agricultural workers. The literature provides a deep knowledge regarding the health burden that agrichemical use entails; however, in regard to mental health, the literature contains conflicting definitions, diagnostic criteria and data and exposes a significant dearth of quality research assessing the associations between agrichemical exposure and poor mental health. It is important to note that research variously explains poor mental health as psychological distress, clinically significant negative emotional states and diagnosed mental health conditions (as determined by the DSM-5) [6]. Despite the variability, each of these constructs helps shape our understanding of mental health and will be taken into consideration when determining the results of this review.

### 1.5. Aims of the Review

Farming lays the foundations of a strong, wealthy and successful society. Neglecting the mental health of farmers will be to the detriment of the growth and transformation of rural communities. TWH™ refers to the multipronged and multi-sectorial approach to providing protection from work-related health and safety issues, concurrently acting in primary and secondary prevention to overall potentiate workers’ wellbeing [53]. This review seeks to: (1) improve the understanding of the links between agrichemical exposure and farmer mental health, (2) gain insight into how chronic (low-level) exposures impact mental health and suicide risk, and (3) understand how this may guide screening, prevention and policy developments in the future.

## 2. Methods

The review was conducted using search strategies on search engines, government documents and websites and utilised key words to meet its aims. The search terms utilized were: psychological distress, farmer health, farmer mental*, rural, Australia* farmer, suicid*, pesticide mental* and organophosphate. These search terms were also used in combination with Boolean operators ‘AND’ and ‘OR’ to increase the scope of the search parameters. PubMed was the main database utilized, with the use of Google Scholar to supplement the reach. Searches were not restricted by the age of the articles to potentiate the depth of the review. Articles were assessed initially on the basis of titles and abstracts for relevance, followed by full-text access to clarify their necessity for inclusion into the review. The following criteria were employed to help curate the search material:Research reported on pesticide effect on mental healthFull text of research article was in EnglishResearch was peer-reviewedResearch focus was on a farming/agricultural contextResearch focusing on intentional exposures to pesticide (used for suicide/homicide) were excludedFocus on organophosphate-induced health effects

Despite not being a systematic review or meta-analysis, the selected publications were reviewed using the PRISMA guidelines (Figure 1; Table 1 and Table 2) [54]. Local Australian research was incorporated but, given the importance of the TWH™ paradigm, international data were also utilised as examples within the review. From here, a snowballing and saturation approach was utilized [55]. Reference lists of appropriate reviews were assessed to streamline and guide the search efforts for relevant research material. Given that this review provides a focus on a little explored area of research, a broad and deep coverage of the topic was of primary concern. Inclusions were based on the initial search criteria, with no exclusions made on the basis of the study type, nor was a limitation placed on the date of publication. Rather, the studies discussed herein were critically evaluated for their strength of evidence and their methodological merit in an overall effort to provide a comprehensive review in this area of research.

## 3. Results

### 3.1. Challenges of Determining Poor Mental Health and Suicide Rates

A broad range of methods have been used to determine poor mental health within the agrichemical exposure literature. Most commonly (N = 24), clinically significant negative emotional states (such as depression and anxiety) were determined using validated assessment tools [55,56,57,58,59,60,61,62,63,64,65,66,67,68,69,70,71,72,73,74,75,76,77,78,79]. These were generally administered via questionnaires (N = 24), either verbally or self-administered using paper-based or online formats [55,56,57,58,59,60,61,62,63,64,65,66,67,68,69,70,71,72,73,74,75,76,77,78,79]. Some studies (N = 5) utilised self-report questions about lifetime diagnosis and treatment for depression [60,61,66,70,71]. A small number of studies (N = 2) included clinical evaluation by a mental health professional using the criteria set by the American Psychological Association’s DSM [72,78].

The time frame over which poor mental health has been measured is also variable across the literature. Assessment time frames ranged from symptoms experienced ‘in the past week’ [69], DSM requirements for symptoms experienced in the past two weeks [72,78], through to a ‘have you *ever* been diagnosed…’ or ‘have you *ever* received treatment…’ [61,66,70,71]. Additionally, various measures equating to poor mental health have been used across the agrichemical exposure literature, including mood [56,60,74,78], depression [58,59,61,62,63,65,66,68,69,70,71,72,76,78], anxiety [57,58,59,65,69,72,76,79], affective psychosis [80], psychological distress/symptoms [64,67,73,75] and suicide attempt [61,80,81,82,83,84,85].

The collection of suicide data across the literature is also likely to be variable, given the wide range of countries included in the data. Previous research has already identified widespread variability in suicide data collection, coding, data management and data reporting—resulting in lack of clarity around accurate farmer suicide statistics [52].

This noted variability makes comparing research methods and outcomes challenging and limits the ability to draw broad conclusions about links between agrichemical exposure and poor mental health outcomes.

### 3.2. The Effect of Organophosphate Exposure on Mental Health

A large body of literature has described the hazardous effect of high-dose and/or short-term exposure to organophosphates [25,86,87,88]. However, the epidemiological data assessing the association with psychological distress are conflicting.

Seminal studies of organophosphates during the mid-1900s gave rise to reports of anxiety and mood disorders [89,90,91,92,93] as well as accounts of depression and schizophrenia post-organophosphate poisoning [90]. The subsequent collection of studies during the mid to late 20th century culminated in an acceptance that depression and anxiety are a sequela of organophosphate exposure. Studies comparing pesticide-exposed cohorts to control populations reported significantly higher anxiety scores [57], even two years after a poisoning event [60]. However, further assessment of the literature questions such clear associations between psychological distress and pesticide exposure (see Table 1).

#### 3.2.1. Epidemiological Evidence Displays Inconsistencies but Suggests a Viable Link

A body of research has developed a focus on the links between agrichemical use and mental health within a range of farming populations. An ecological study showed farmers in areas of high pesticide use were at greater risk of affective disorders compared to control populations [82]. A study in banana workers in Costa Rica identified high-dose pesticide poisoning was significantly associated with higher scores for anxiety, depressive symptoms and personality disorders [64,94]. Intriguingly, the study also suggested for the first time a statistically significant dose–response effect may exist, such that psychological distress symptom scores increased with the number of past poisoning events, with statistical significance [64,94]. A study in sheep farmers described higher symptoms of depression in exposed sheep farmers compared to unexposed rural workers [58]. However, some of these studies failed to control for known risk factors for affective disorders such as socioeconomic status, age, substance use and, moreover, used small cohort sizes. One prospective study showed depressive symptoms in those with past pesticide poisoning but also failed to account for confounders [62]. To further highlight the discrepancy in the literature, a separate ecological study by Parron and colleagues [80] did not find higher rates of affective disorders for those living in an area of high pesticide use compared to those living in low-exposure regions (Table 1).

There are recent suggestions that the relationship between depression and agrichemicals is dependent upon the severity of exposure. This stems from a study that assessed cholinesterase levels, as a marker of level of poisoning, compared to symptoms of depression [95]. The results were similar to those of a recent study in South Korea [63]; however, both relied on self-reports of depression rather than on clinical diagnoses. Further, both studies only utilised a marker of poisoning (cholinesterase levels), symptomology, or the number of poisoning events as a gauge to the severity of poisoning. Therefore, it cannot be concluded that the severity of poisoning dictates the development of psychological distress, either in the short or in the long term.

A number of other studies also failed to establish any associations [66,67,69]. This may be explained by the small sample sizes, focus of studies (occupational group studied), methodological use (exposure rates/routes/duration, protective clothing, self-reporting) and interpretations between studies [25,96]. As an example, a study by Delgado and colleagues described that psychiatric symptoms reported by participants with acute poisoning were found to increase over a two-year period compared to controls. However, once confounders were applied, the difference between the study groups was not significant [67].

With respect to study design, most studies in this field utilise cross-sectional and ecological designs (Table 1), both of which only allow the assessment of associations but not cause–effect relationships. Epidemiological studies overall are also hampered by potential for recall bias, particularly in cross-sectional modalities. Of the studies that have assessed the association between mental health and toxic pesticide exposure, only a small sample involved longitudinal evaluations [59,65,68,71] (Table 2).

#### 3.2.2. Stringent Studies Lay Foundations for the Association between Agrichemical Exposure and Poor Mental Health

A large U.S. agricultural health study demonstrated the strongest associations between psychological distress and pesticide exposure (Table 2). The study involved farm workers and their spouses, finding that the women in the study with prior pesticide poisoning were at three times greater risk of depression [70]. A subsequent study from the same group identified that both high- level and low-level pesticide exposure were associated with clinically diagnosed depression [71]. A significant study by Beard and colleagues assessed for relationships between 10 functional classes of pesticides and diagnosed mood disorders in a cohort of male pesticide applicators [66]. They found a positive association between exposure to pesticides, including organophosphate-based applications, and depression [66]. Intriguingly, they demonstrated this was independent of pesticide poisoning. These studies provide a more stringent analysis, since they controlled for other known risk factors such as mood disorders [62,70,71,96].

Mackenzie Ross and colleagues utilised structured clinical interviews to assess psychological distress in a cohort of sheep farmers. Their earlier studies showed high prevalence of anxiety and depression in those exposed to pesticides on the basis of self-reporting (accounting for other variables) [58,72]. Their diagnostically rigorous follow-up study only demonstrated a correlation with clinically diagnosed anxiety [72]. Overall, the extent to which chronic low-dose exposure to pesticide can influence psychological distress is unclear.

#### 3.2.3. Unclear Association between Agrichemical Use and Suicidality

The literature confirms that environmental or occupational exposures can disturb neurochemistry and, hence, predispose to psychological distress [97]. However, whether suicidality is influenced by organophosphate exposure is of significant interest and remains unknown. This is of particular concern, given heightened suicide rates in farming populations [14,15,19,25].

The use of pesticides for acts of self-poisoning is well reported. However, associations between pesticide exposure and suicide risk have only been highlighted in the literature within the last 25–30 years. A seminal study amongst forestry workers identified increased rates of suicide from occupational exposure to pesticides [81] (Table 2). Similar findings were identified in cross-sectional studies among banana workers with past history of organophosphate poisoning [77] and Chinese workers who stored pesticides at home [85], showing greater suicidal ideation compared to control participants. Meyer and colleagues described young adult participants (20–39 years of age) exposed to intense levels of agrichemicals had higher risk of suicide compared to other populations [82]. Similar studies described the same result even if to a lesser effect [83], but all failed to control for other risk factors for suicide [82,83,84].

In contrast to these studies above, a recent prospective study found pesticide use did not increase suicide risk [66], thus underscoring the current inconsistencies in the field. There has been a dearth of well-designed research studies to explore these links in detail. Therefore, valid conclusions cannot yet be drawn.

**Table 1 ijerph-16-01327-t001:** Characteristics of the studies published on pesticide exposure and effects on neurobehavioural or psychiatric disturbances in those in agricultural occupations.

Study and Year	Study Design	Region	Population Source	Exposure Interest	Outcome Focus	Outcome Measure	Results *
Salvi et al., 2003 [59]	Longitudinal	Brazil	Agricultural tobacco farmers 62 participants	Organophosphate exposure	Neuropsychological (extrapyramidal symptoms), psychiatric (depression, anxiety)	MINI – Mini-International Neuropsychiatric Interview (structured questionnaire administered by a psychiatrist)	Three months of organophosphate-free period reduced diagnoses of psychiatric diagnoses
Fiedler et al., 1997 [56]	Cross-sectional	USA	Fruit farmers 99 participants	Organophosphate exposure	Neuropsychological	Minnesota Multiphasic Personality Inventory (MMPI-2)	No significant differences in mood
Weisskopf et al., 2013 [61]	Cross-sectional	France	Agricultural workers 781 participants	Pesticide exposure	Depression	Single question asking whether they had ever been treated with antidepressants, lithium or sismotherapy, or hospitalised for depression	Elevated depression rate in those using herbicides. Dose–response relationship identified for duration and intensity of use.
Mackenzie Ross et al., 2010 [58]	Cross-sectional	England	Sheep dippers 205 participants	Organophosphate exposure	Anxiety, Depression	Hospital Anxiety and Depression Scale	Anxiety and depression higher in exposed group (40% of exposed vs 23% of controls
Levin et al., 1976 [57]	Cross-sectional	USA	Commercial pesticide sprayers	Organophosphate exposure	Anxiety	Taylor Manifest Anxiety Scale (derived from the MMPI), Beck Depression Inventory	Sprayers showed higher anxiety levels and lower acetylcholinesterase (AChE) levels compared to controls, no difference in depression scores
Savage et al., 1988 [60]	Case-control	USA	Pesticide applicators 100 participants	Organophosphate poisoning	Neurobehavioural (memory, abstraction, reflexes), Mood	MMPI	Those with past poisoning had intellectual function scores consistent with individuals with cerebral damage or dysfunction.
Meyer et al., 2010 [82]	Ecological	Brazil (Rio)	Agricultural workers 3517 participants	Use of pesticides	Hospitalisation due to suicide attempts 1998–2007. Suicide deaths in 1981–2005.	Suicide deaths from the Brazilian National Mortality System (using WHO International Classification of Diseases and Related Health Problems (ICD). Hospitalisations due to suicide attempts or mood disorders from the Brazilian Hospital Information System (using ICD)	Suicide: agricultural workers at higher suicide mortality risk compared to three reference populations. Hospitalisation: Higher rates following suicide attempts/mood disorders, also compared to reference populations.
Wesseling et al., 2002 [64]	Cross-Sectional	Costa Rica	Banana planation 211 participants	Reduction in pesticide exposure	Neuropsychiatric symptoms	Questionnaire-16 and Brief Symptom Inventory (BSI)	Marked increase in neuropsychiatric symptoms observed in organophosphate-poisoned workers compared to controls
Beseler and Stallones, 2008 [62]	Prospective/Longitudinal	USA (Colorado)	Farm residents and spouses: CFFHHS Project 653 participants	Pesticide poisoning at baseline (1993): ever or never	Depressive symptoms: CES-D scale	Center for Epidemiologic Studies-Depression scale	Symptoms of depression were associated with participants that had a history of pesticide poisoning.
Parron et al., 2011 [80]	Ecological	Spain (Andalusia)	General population with neurological disorders 1349 participants	High vs Low pesticide exposure areas	Affective psychosis, Suicide attempts.	Hospital records (Andalusian Health Service Minimum Dataset)	Rates and risk of suicide and affective disorders found to be higher in populations exposed to higher levels of pesticides compared to populations exposed to lower levels
Kim et al., 2013 [63]	Cross-sectional	South Korea	Male farmers 1958 participants	Pesticide poisoning	Depressive symptoms	Face-to-face administering of the Korean version of the Geriatric Depression Screening Scale (short form)	Risk of depressive symptoms increased with pesticide poisoning (OR - 1.61, 95% CI, 1.10–2.34). Risk increased with severity of poisoning symptoms.
Beard et al., 2014 [66]	Cross-sectional	USA (Iowa and North Carolina)	Agricultural Health Study	Pesticide exposure	Depression	Single written questionnaire question: “Has a DOCTOR ever told you that you had (been diagnosed with) depression requiring medication or shock therapy?”	Positive association between depression and occupational pesticide use among applicators
Solomon et al., 2007 [69]	Cross-sectional	UK	Sheep dippers 9844 participants	Pesticide exposure	Anxiety, Depression	Written questionnaire including questions from the Hospital Anxiety and Depression Scale (symptoms experienced in the past 7 days)	Past use of pesticides not associated with anxiety and depression.
Delgado et al., 2004 [67]	Prospective	Nicaragua	Hospitalised patients from pesticide poisoning 81 participants	Organophosphate poisoning	Psychiatric symptoms	Modified Spanish version of the Q-16 assessing neuropsychiatric symptoms	Psychiatric symptoms increase with time since the poisoning event.
Bazylewicz-Walczak et al., 1999 [65]	Cross-sectional	Poland	Greenhouse workers and unexposed controls 51 participants	Organophosphate exposure	Depression and anxiety questionnaires before and after spraying season	Subclinical neurobehavioural effects using the World Health Organization (WHO) Neurobehavioral Core Test Battery (NCTB)	Increased anxiety, anger, fatigue, depression symptoms. N.B. No significant effects of exposure after a single spraying season.
Onwuameze et al., 2013 [68]	Longitudinal	USA (Iowa)	Iowa Certified Safe Farm study	Pesticide exposure	Self-reported depressive symptoms	Single written questionnaire asked quarterly throughout study: “How would you rate your level of depression in the last quarter?”	Pesticide exposure prospectively increased risk of depressive symptoms
Beseler et al., 2008 [71]	Nested case-control	USA (Iowa and North Carolina)	Agricultural Health Study 15,585 participants	Cumulative pesticide exposure: <226 days (low), 226–752 days (intermediate), >752 days (high). Diagnosed pesticide poisoning	Self-reported or medically diagnosed depression	Single written questionnaire question: “Has a DOCTOR ever told you that you had (been diagnosed with) depression requiring medication or shock therapy?”	Pesticide poisoning more strongly associated with depression than high cumulative exposure. However, high cumulative exposure in the absence of poisoning significantly associated with depression
Beseler et al., 2006 [70]	Nested case-control	USA (Iowa and North Carolina)	Agricultural Health Study 29,704 participants	Pesticide exposure	Self-reported or medically diagnosed depression	Single written questionnaire question: “Has a DOCTOR ever told you that you had (been diagnosed with) depression requiring medication or shock therapy?”	Depression significantly associated with history of pesticide poisoning but not with low or cumulative exposure
Harrison et al. 2016 [72]	Cross-sectional	England	Sheep farmers 205 participants	Organophosphate exposure	Anxiety and depression	Hospital Anxiety and Depression Scale, Beck Anxiety and Depression Inventories, Structured Clinical Interview (DSM-IV criteria)	Exposed cohort reported higher rates of depression and anxiety but only held true for anxiety when diagnostic interviews were utilised

* Neurobiological studies were included here, as these studies reported mental health outcomes as part of neurobiology and/or neurobehavioural findings.

**Table 2 ijerph-16-01327-t002:** Characteristics of the studies published on the effects of low-dose pesticide exposure on mental health outcomes and the effects of pesticide exposure on suicidality.

Study and Year	Study Design	Region	Population Source	Exposure Interest	Outcome Focus	Outcome Measure	Results
Stephens et al., 1995 [73]	Cross-sectional	England	Sheep dippers 289 participants	Pesticide exposure	Neurobehavioral	General Health Questionnaire screening tool	Increased susceptibility to psychiatric disorders.
Farahat et al., 2003 [75]	Cross-sectional	Egypt	Cotton crop *workers* 102 participants	Organophosphate exposure	Neurobehavioral	Eysenck Personality Assessment Questionnaire (EPQ)	Statistically significant lower performance in neurobehavioral test for the exposed group. The longer exposure correlated with worsening performance.
Green et al., 1991 [81]	Cohort	Canada	Forestry workers 1222 participants	Phenoxy acid herbicides	Suicide/mortality	Canadian Mortality Database using ICD codes	Statistically significant increase in deaths from suicide for exposed cohort.
Wesseling et al., 2010 [77]	Cross-sectional	Costa Rica	Banana plantation 208 participants	Previous poisoning with cholinesterase-inhibiting pesticide	Suicidal ideation	Verbal administration of BSI – measuring psychological distress	Higher prevalence of somatisation, depression and anxiety disorders. Odds ratio for suicidal thoughts was 3.72.
Zhang et al., 2009 [85]	Cross-sectional	China	Rural residents 9811 participants	Storage of pesticide	Suicidal ideation	Verbal administration of questions to ascertain whether the respondent reported suicidal ideation in the two years before the interview.	Increased rates of suicide for cohort using pesticides. Also, higher incidence of depression and neuropsychiatric disturbances.
Parron et al., 1996 [83]	Ecological	Spain (Andalusia)	Agricultural workers 251 participants	Pesticide exposure	Suicide deaths	Suicides determined as cause of death (source of records not stated), psychological autopsy method used to determine reasons for suicide	Increased rates of suicide using pesticides. Higher incidence of depression and neuropsychiatric disturbances.
Pires et al., 2005 [84]	Ecological	Brazil	Agricultural workers 640 participants	Insecticides and herbicides	Suicide attempts and death	Suicide attempts determined from records of the Integrated Center for Toxicological Surveillance of the State Health Department of the State of Mato Grosso do Sul. Suicide deaths determined using data from Epidemiology Division of the State Health Secretariat of Mato Grosso do Sul using ICD-10 codes	Increased prevalence of suicide attempts and deaths in regions where higher insecticide use occurred compared with areas of lower use.
Ames et al., 1995 [74]	Cross-sectional	USA	Agricultural workers 135 participants	Organophosphate exposure	Mood	Mood scales from the Neurobehavioral Evaluation System (computerized assessment)	No significant abnormality in mood.
Jamal et al., 2002 [76]	Case-control	England	Sheep farmers 72 participants	Organophosphate exposure	Neuropathy, Depression and anxiety	Mood and affect assessed using General Health Questionnaire and the Hospital Anxiety and Depression Test	Increasing neuropathy associated with exposure. Neuropathy correlated to increased anxiety and depression scores
Amr et al., 1997 [78]	Cross-sectional	Egypt	Urban textile workers 503 participants	Pesticide exposure of formulators and applicators of pesticides	Depressive disorders and mood symptoms (insomnia, anhedonia, anxiety)	In-field screening using General Health Questionnaire, further in-field diagnosis by a psychiatrist according to DSM-III-R criteria	Higher depression in exposed subjects over controls. Particularly, those with longer-term exposure (> 20 years)
Roldan-Tapia et al., 2006 [79]	Cross-sectional	Spain	Greenhouse workers 92 participants	Carbamate and organophosphate poisoning	Neuropsychological performance	Taylor Anxiety Scale & Beck Depression Inventory	Exposure linked to increased anxiety. However, exposure for less than 10 years associated with profiles similar to unexposed controls.

### 3.3. Lack of Data on Chronic Low-Level Exposure of Agrichemicals

Perhaps the greatest deficiency of research in this field is the dearth of knowledge about the effect of chronic low-level (or asymptomatic) exposure to agrichemicals on mental health. Very few studies to date have directly evaluated the effect of chronic exposure to low levels of pesticides, in the absence of poisoning, on the development of psychological distress.

Limited studies have attempted to compare exposed and non-exposed agricultural workers. One identified higher anxiety levels in exposed farmers but was hampered by small sample sizes and unclear history of assumed non-exposed farmers [98]. Yet, a similar study failed to identify variance in neurobehavioural outcomes in exposed subjects; however, the conclusions are stymied by poorly matched control subjects (age, education) as well as unclear duration of exposure [74]. Roldan-Tapia (2006) and colleagues studied a cohort of pesticide applicators with an average exposure history of 10 years [79]. They reported higher anxiety levels in pesticide applicators compared to controls, as well as delays in higher level neurobehavioural testing, but the assessment of psychological distress was based upon surveys and not clinical evaluations [79].

### 3.4. Variability in Findings IS Linked to Inconsistencies in Study Designs

The literature is stymied by inconsistencies in the diagnosis, reporting and recording of mental illness [25]. A number of studies utilise self-report mental health surveys with varying levels of consistency [74,75,76], while others base conclusions on a single-item assessment, e.g., one question on suicidal thoughts [77]. Clinical interviews conducted by a mental health professional are considered the gold standard in diagnosing or simply even identifying mental health concerns. A study of Egyptian farmers used clinical interviews to diagnose participants according to the DSM [78]. Although mental illness was found in those exposed to pesticides, little was known about exposure histories. Therefore, the findings could be variously due to long-term asymptomatic exposure or past history of high-level acute poisoning.

Another confounding factor in past research is the lack of accounting for other known determinants or risk factors for mental disorders. As eluded to previously, socioeconomic factors in a rural or farming setting can play a significant role in psychological distress [13,31] and are rarely considered in findings. Furthermore, a lack of detail in participants’ psychological history clouds the understanding of whether pre-existing mental health issues or the chemical exposure accounts for positive associations described in the study. Given the complexity of the subject area, many researchers have focused on assessing prevalence rates of mental illness between exposed farmers compared to control subjects [72,74].

## 4. Discussion

### 4.1. Key Findings and Implications for Policy and Practice

This review set out to understand the effects of agrichemical exposures on the mental health of famers. Overall, the review does suggest a tangible link between mental health outcomes and pesticide exposure. Particularly pertinent to this study, there does appear to be an association between poor mental health and the previously underreported area of chronic low-dose exposure to pesticide. This is an important finding for a multitude of reasons, not least that this pattern of exposure may occur on a frequent basis in agricultural workforces, even in those observing current best practice safety and pesticide use techniques. Further, this underscores the need to be proactive and protect the health of agricultural workers worldwide using a public health approach. Protecting the health of farming populations requires a combination of education and practice, policy and regulatory change.

The literature reports that education on safe handling and usage of agrichemicals, as well as the harmful effects of exposure can reduce workplace exposures to pesticides [99,100]. However, this needs to be contextually based and targeted, as some research describes educational interventions in farmers are not always effective in preventing exposures and harm [101]. Education also needs to be broadened to address both chronic low-dose exposure and acute exposure.

Influencing change in agrichemical usage and behaviours remains challenging, particularly when family farming remains the dominant business arrangement, making enforced regulatory change not a viable or palatable option. This is more easily achieved in a corporatized model, where usually policy, procedures and workplace behaviour expectation are clearly documented. While creating a positive and enduring behavioural change is challenging, it should not be viewed as impossible. A successful example can be seen in the Sustainable Farm Families™ (SFF) program delivered across Australia and more recently in Alberta, Canada [39]. This program challenged the traditional view that farmers are not interested in their health by developing a program that reflected an understanding of life and work in the farming context. The SFF model of delivery emphasised the significance of farmer health, wellbeing and safety on the success of the farming business. It also showed significant improvements across measures of health, wellbeing and safety practices of farmers [102,103]. Ensuring that agrichemical education is relevant and meaningful to farming populations and relates to self- and family success, as well as to business profitability, is imperative for stimulating practice and culture changes associated with agrichemical usage in farming communities [104]. A similar successful public health initiative exists in the U.S. (Iowa Certified Safe Farm study) [68].

It is not realistic to think that we can eliminate the use of all agrichemicals. Policy-makers should be directed to reduce access to and restrict the use of the more hazardous of chemicals—particularly in developing nations. However, protection from the hazards of agrichemical exposure—when their use is unavoidable—must also be considered by the users. The need to develop effective, efficient and cost-effective personal and protective equipment is a global issue. Whereas farmers in developed countries generally have access to commercially available personal protective equipment, in addition to lower exposures due to better engineered products and machinery, this is not the reality for farming populations in developing countries, where basic harm minimisation techniques can make a big difference.

Education and practice change should be encouraged not only in farming populations but also in those health and rural professionals who support them. Developing cultural competency across rural health professionals will help identify potential risk factors and situations faced by their clients and assist with accurate diagnosis, treatment and prevention from further or ongoing risk to mental health [105].

### 4.2. Limitations of the Existing Literature

Of particular concern in the literature to date is the lack of clarity in classifying pesticide exposure. In the past, the literature has used the descriptors ‘long-term exposure’ and ‘acute poisoning’ synonymously. While acute poisoning can have strong detrimental health effects that warrant greater understanding, it is now suggested that there are detrimental mental health effects from sub-toxic levels of pesticide in the long term. Future research should provide case definitions to complement the varying effects of pesticide exposure. This involves recognising that *long-term exposure* or *chronic exposure* should now be heralded as an umbrella term to encompass all modalities of chronic exposure. Under this, the subcategories should include *chronic toxicity*, referring to exposures to toxic events in the past, and the independent definition of *chronic low-dose exposure* to facilitate appropriate analysis, discussion and context.

Further challenges identified within the literature to date included the use of the descriptor “neurobehavioral” synonymously with mental health conditions. Additionally, the outcomes of research at times included a combination of neurobehavioral symptoms and mental health conditions, making identifying and isolating only mental health conditions challenging.

### 4.3. Recommendations for Future Research

A key component of an epidemiological attempt in exploring the association between a chemical and mental illness is a reliable measure of exposure. Lack of such a measure has hindered research to date. Characterising and assessing pesticide exposure is complex and requires a stringent methodology to ensure quality long-term exposure data [106,107]. Typically, the most ardent indicator involves those quantifying exposure [108], such that dose—esponse relationships can establish a causal relationship. There are currently several methods for measuring exposure, but they present limitations when applied in the field. Self-report of exposure is easy to obtain but often affected by memory and bias and are subject to unreliability [109]. Some farming cultures may create power imbalances between farm workers and farm owners, thereby misrepresenting or skewing data collection. Validated questionnaires can help curb such issues but, overall, still represent a blunt tool. Biomarker techniques represents another powerful method of exposure measurement, as they account for various routes of absorption and means of exposure, although are expensive [109].

A significant advancement in research would be the implementation of longitudinal studies and, more specifically, prospective studies [110]. This would provide ideal settings to assess mental health prior to exposure of organophosphates and establish the pre-morbid state. Subsequent tracking and assessment of exposure events with follow-up mental health assessments upon cessation of organophosphate use will ultimately yield more conclusive data. The inclusion of biological measures of exposure would further help to delineate any existing dose–response relationship with a view to establishing any causal relationship between low-level exposure to pesticide and psychological disturbances. The feasibility of such an undertaking requires available funding.

Of great importance in future studies is the consideration of potential confounders and accounting for other mental health stressors apart from pesticide exposure. The relevance is underpinned when discussing pesticide exposure and its link to suicidality. Farmers are undoubtedly exposed to a plethora of conditions and practices that are potentially injurious to health and are inextricably connected to mental wellbeing. The impact on health may in turn lead to higher risk of substance use, secondary health outcomes and, potentially, suicide [111]. Moreover, study populations may be impacted by poor mental health in different ways. As an example, studies in developing nations include participants compounded by low income, low education, poor availability and access to healthcare, and very low health literacy. This constellation of factors may contribute to increased susceptibility to the neurological effects of pesticide exposure and, hence, a higher rate and/or steeper pathway towards suicidality.

Given the sheer levels of complexity in this field, one must be open to the possibility of complex explanations. Although some research has sought to establish a causal relationship between psychological distress and pesticide exposure, this relationship may not be so simple. As an example, an increasing number of studies within this field have focused on elucidating the activity of the enzyme paraoxonase (PON1) with respect to organophosphate exposure. As a liver enzyme, it contributes to the metabolism of organophosphates, however, PON1 mutations and low PON1 activity levels have been found in those reporting poor health [112,113]. Therefore, participants who report poor health more often are also those less efficient at metabolising organophosphate. It is nonetheless feasible that those who are more susceptible to ill health may also suffer from psychological distress. Being open to this level of complexity may allow more directed studies to develop in the future.

## 5. Conclusions

Exposure to organophosphates and outcomes impacting psychological distress, mental health and suicide risk need further research [87]. The mental health literature supports that environmental or occupational exposures can disturb neurochemistry and, hence, predispose to psychological distress [97]. Despite inconsistencies in this field, there is no doubt that agrichemical use needs to be regulated to protect farmers’ mental health in the future. Further developments in the field will help inform public health policies regarding users’ education, safe exposure levels, appropriate personal protective equipment, targeted screening and treatment as dictated by the TWH model [34,35,53]. It is also possible that a further restriction of pesticide use may also reduce total exposures globally.

## Figures and Tables

**Figure 1 ijerph-16-01327-f001:**
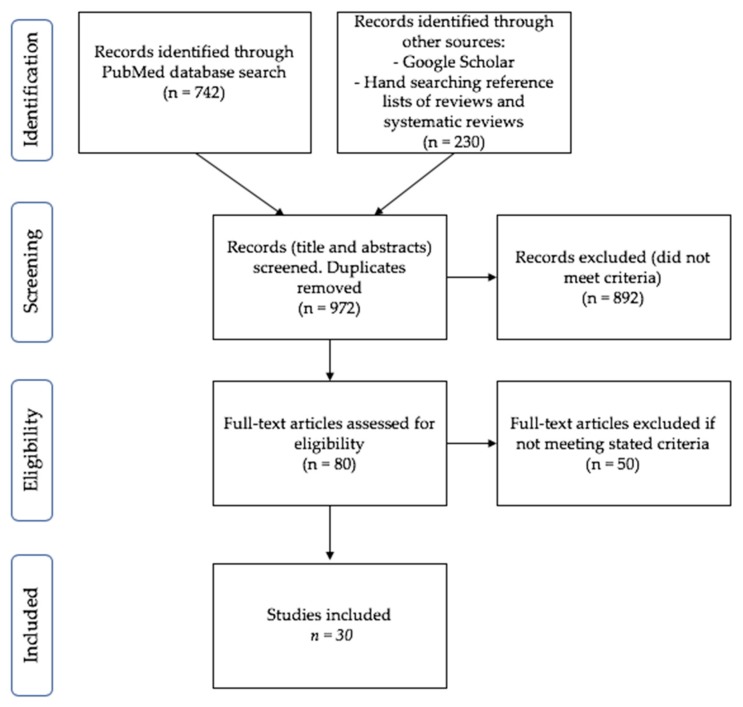
PRISMA diagram of research publication selection.

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
