# Peer review of "A Pest to Mental Health? Exploring the Link between Exposure to Agrichemicals in Farmers and Mental Health"

_ijerph, 2019, doi:10.3390/ijerph16081327_

Reviewer 1 Report

General comment

The paper “A Pest to Mental Health? Exploring the Link between Exposure to Agrichemicals in Farmers and Psychological Distress” explores the interesting, and not fully explained, link between the exposure to pesticides, other agrochemicals, and farmers’ mental health. It is a well-written paper.  Unfortunately, the methodology used in this review paper is not reproducible and lacks many details as explained below. In addition, the Results report too many information, which do not fall under any of the two aims underlined in Section 1.6 of the paper. The information provided is in any case interesting and warrants publication, but should be more focused on the title and aims at hand. I suggest shortening the paper by excluding unnecessary information or condensing the specific information the Authors find useful. For example, the real information, which falls under the aim of this study in the Results, starts only in Section 3.6.

Specific comments

Line 12: The Abstract should also contain some information about the methods used for the review, some of the main findings, and conclusions so that a person can understand what the paper brings even without reading the full paper.

Line 25: I am not fully aware with the messages (conclusions?) found as sub-headings under main headings.

Line 38: I do not understand what are “subpopulations of the wider community”. It does not become clear before the last sentence in the paragraph, therefore I suggest rephrasing.

Lines 145-160: Since this is a review article, it would be much more useful for the readers to understand clearly the search strategy used. “Various combinations of words…” does not make the methodology of this paper reproducible. The Authors should state clearly what search engines were used (all of them), what search strategy/string was used with which engine, as well as what were the exclusion criteria – since there was an initial assessment and it would be interesting to know how many abstracts were found initially, how and why some were excluded and other included, etc…

Line 163: The point 3.1 of the Results does not cover any of the aims of the paper, except in the last paragraph where they seem to have been mentioned only vaguely.

Line 201: The Authors state that “intentional exposures… go beyond the scope of this review” but this is not underlined in the Methods section of the paper. This is why a more detailed Method section is necessary.

Results in general should be much more focused on the title and aim of the paper. The same goes for the Discussion and Conclusions. A person searching for the information promised in the Title of this paper would find it difficult to draw any conclusions.

Author Response

Dear Editor,

Re: A Pest to Mental Health? Exploring the Link between Exposure to Agrichemicals in Farmers and Mental Health

Thank you for your thorough review and considered responses to our original manuscript submission. In light of this feedback, we have made significant changes to the manuscript and feel that it will provide a considerably enhanced article pertinent to the readership of your journal. I would like to re-submit the revised paper with specific changes as suggested by you and the peer-reviewers.  

Please find the following changes in the revised manuscript as requested:

-       An item-by-item response to reviewer comments below.

-       Changes have been tracked (note, deletions which have been accepted have unfortunately not remained highlighted in the tracked document, but hopefully the item-by-item response helps clarify our alterations)

Once again thank you for your constructive comments and we trust the revised manuscript meets your requirements. 

Yours sincerely,

Dr. Nufail Khan

Responses to the reviewer 

Reviewer 1 response (RR1):

RR1:
Line 12: The Abstract should also contain some information about the methods used for the review, some of the main findings, and conclusions so that a person can understand what the paper brings even without reading the full paper.

Response:

The abstract has been completely re-written with the comments in mind

RR1:
Line 25: I am not fully aware with the messages (conclusions?) found as sub-headings under main headings.

Response:

The title has been shortened and adjusted to be more applicable to the introduction. Furthermore, the introduction itself has been re-formatted with removal of section 1.2 and significant editing of other sub-sections. 

RR1:
Line 38: I do not understand what are “subpopulations of the wider community”. It does not become clear before the last sentence in the paragraph, therefore I suggest rephrasing.

Response:

The paragraph which began at line 40 has been re-written and re-formatted to provide a clearer explanation of populations affected by mental illness disproportionaltely, to allow discussion of agriculture in the following paragraph and sections.

RR1:
Lines 145-160: Since this is a review article, it would be much more useful for the readers to understand clearly the search strategy used. “Various combinations of words…” does not make the methodology of this paper reproducible. The Authors should state clearly what search engines were used (all of them), what search strategy/string was used with which engine, as well as what were the exclusion criteria – since there was an initial assessment and it would be interesting to know how many abstracts were found initially, how and why some were excluded and other included, etc…

Response:

The methods section has been altered drastically. Following editing of the initial text, we have now added the criteria used to curate the search material. Furthermore, even though this isn’t a systematic review and we have maintained this piece as such, a PRISMA flow diagram has been produced and inserted as figure 1 in the methods section. Finally, two tables displaying the characteristics of the 33 publications utilised in this study have been constructed and inserted as Table 1 and 2 in the methods. We believe this helps to ensure reproducibility of our work.

RR1:
Line 163: The point 3.1 of the Results does not cover any of the aims of the paper, except in the last paragraph where they seem to have been mentioned only vaguely.

Response:

We agree with this comment and hence we have edited 3.1 signidficantly. We also believe it holds relevance in setting the piece for the subsequent analytical section.

RR1: 
Line 201: The Authors state that “intentional exposures… go beyond the scope of this review” but this is not underlined in the Methods section of the paper. This is why a more detailed Method section is necessary.

Response:

Within the methods section one of the criteria for research selection is listed as “research focusing on intentional exposures to pesticides (used for suicide/homicide) were excluded.” Furthermore we have removed the stated text from results section to further allow directed discussion as requested by the reviewers.

RR1: 
Results in general should be much more focused on the title and aim of the paper. The same goes for the Discussion and Conclusions. A person searching for the information promised in the Title of this paper would find it difficult to draw any conclusions.

Response:

The authors have taken this under real consideration. As such the results section has been heavily modified. Firstly the aims of the paper have been used to help create the subheadings in the results section i.e. 3.1 is focused on the “Current understandings of agrichemical exposure” whislt 3.2 is focused on the “affect of organophosphate exposure on psychological distress and mental health.” In this way it forces the results section to be aligned with the aims throughout.

Furthermore, prior to re-adjusment of headings, section 3.3 has been removed in its entirety. Section 3.4 was split into subsections, edited in its content, and combined with section 3.5.

The discussion section has also been heavily formatted with removal of two paragraphs in its entirety. There is addition of a new paragraph at the end of the section to expand on the discussion about how this research may aid development of public health policies using examples from Australia.

Finally, the conlusion is edited to be extremely concise and directed towards the aims.

Reviewer 2 Report

This paper addresses relevant topic, that is worthwhile to research.

However, in order to meet scientific standards and deliver quality research with more valid and reliable evidence, I would suggest authors to conduct a systematic literature review and design the manuscript accordingly.

The following publications might be helpful:

Freire, C., & Koifman, S. (2013). Pesticides, depression and suicide: a systematic review of the epidemiological evidence. International journal of hygiene and environmental health, 216(4), 445-460.

Moher, D., Liberati, A., Tetzlaff, J., & Altman, D. G. (2009). Preferred reporting items for systematic reviews and meta-analyses: the PRISMA statement. Annals of internal medicine, 151(4), 264-269.

This paper addresses an interesting and relevant topic of agrichemical exposure and farmers’ mental health. However, the manuscript needs considerable methodological strengthening, so the more general valid conclusions could be made. Please see my comments as following:

Introduction 

Introduction is quite comprehensive and uses different terminology. I would suggest to shorten it and focus more on farming and mental health (section 1.3.), as oppose to farming and health (section 1.2).

In the section 1.3 there are numerous references on farmers’ suicide rates. Is that focus of this paper or generally farmers’ poor mental health ie. mental health conditions?

Section 1.5 disrupts the flow of the introduction ie. seems disconnected from the previous sections. I suggest the authors associate agrichemical exposure and its potential for detrimental health effects with mental health, by providing more evidence on association of agrichemical exposure and mental health. In addition, the authors could provide more sound rationale of their paper in this section.

 Furthermore, terminology should be consistent, ie. I suggest authors decide between “mental health conditions”, “mental disorders” and “mental illness”.

Aim of the study

Aims of the study are not in line with reported results. Aims of the study should be clear and straightforward.

Methods

The authors should consider developing the systematic search string and carrying out a systematic review. Grey literature sources might be considered as well.

Methodological check of included papers should be included in methods.

The following publications might be helpful in providing information on how to conduct systematic literature review:

·        Freire, C., & Koifman, S. (2013). Pesticides, depression and suicide: a systematic review of the epidemiological evidence. International journal of hygiene and environmental health, 216(4), 445-460.

·        Moher, D., Liberati, A., Tetzlaff, J.,& Altman, D. G. (2009). Preferred reporting items for systematic reviews and meta-analyses: the PRISMA statement. Annals of internal medicine, 151(4), 264-269.

Results

Result section is too comprehensive and not completely in line with aims of this study. For instance, Section 3.1. focuses on organophosphate pesticides in poor health outcomes, which includes much more than mental health. The same is valid for some other subsections. Aims of this study focus on mental health outcomes. Therefore, results should be reported in a way that they correspond to the aim of this study.

Discussion

Similar to Result section, discussion is very broad and general, drifting away from the aims of this study. In my opinion, it should be revised substantially and focus on the aims of this study, i.e. “The piece herein seeks to 1) 138 investigate the current understanding of agrichemical exposure particularly in the less studied area 139 of farmer mental health, 2) and to gain insight into how chronic (low-level) exposures impinge on 140 psychological distress, to potentially guide screening, prevention and policy developments going 141 forward.”

By carrying out systematic literature review with clear outcome measures, authors will have the opportunity to structure their result and discussion sections in a clear manner, meet scientific standards and deliver quality research with more valid and reliable evidence.

Author Response

Dear Editor,

Re: A Pest to Mental Health? Exploring the Link between Exposure to Agrichemicals in Farmers and Mental Health

Thank you for your thorough review and considered responses to our original manuscript submission. In light of this feedback, we have made significant changes to the manuscript and feel that it will provide a considerably enhanced article pertinent to the readership of your journal. I would like to re-submit the revised paper with specific changes as suggested by you and the peer-reviewers.  

Please find the following changes in the revised manuscript as requested:

-       An item-by-item response to reviewer comments below.

-       Changes have been tracked (note, deletions which have been accepted have unfortunately not remained highlighted in the tracked document, but hopefully the item-by-item response helps clarify our alterations)

Once again thank you for your constructive comments and we trust the revised manuscript meets your requirements. 

Yours sincerely,

Dr. Nufail Khan

 Responses to the reviewer 

Reviewer 2 response (RR2):

RR2:
However, in order to meet scientific standards and deliver quality research with more valid and reliable evidence, I would suggest authors to conduct a systematic literature review and design the manuscript accordingly.

Response:

The authors do not feel it necessary to conduct a systematic review. This piece was developed as a scoping review in a broad manner given the lack of research in this field. Itis a topic which draws on a broad range of fields including from the areas of agriculture, mental health, occupational health, and social/behavioural science, and this does not lend itself to a systematic review. Instead, it's important to screen a vast range of literature across these fields to gain an appreciation of the topic. 

Nonetheless, even though this isn’t a systematic review, a PRISMA flow diagram has been produced and inserted as figure 1 in the methods section. Finally, two tables displaying the characteristics of the 33 publications utilised in this study have been constructed and inserted as Table 1 and 2 in the methods.

RR2: 

Introduction  

Introduction is quite comprehensive and uses different terminology. I would suggest to shorten it and focus more on farming and mental health (section 1.3.), as oppose to farming and health (section 1.2).

In the section 1.3 there are numerous references on farmers’ suicide rates. Is that focus of this paper or generally farmers’ poor mental health ie. mental health conditions?

Section 1.5 disrupts the flow of the introduction ie. seems disconnected from the previous sections. I suggest the authors associate agrichemical exposure and its potential for detrimental health effects with mental health, by providing more evidence on association of agrichemical exposure and mental health. In addition, the authors could provide more sound rationale of their paper in this section.

 Furthermore, terminology should be consistent, ie. I suggest authors decide between “mental health conditions”, “mental disorders” and “mental illness”.

Response:

The title has been shortened and adjusted to be more applicable to the introduction. Furthermore, the introduction itself has been re-formatted with removal of section 1.2 and significant editing of other sub-sections. Section 1.3 has been edited to allow a focus on mental health but including suicide as an dichotomous outcome in non-farming populations, to help highlight the unique factors involved in farming i.e. pesticide exposure.

Section 1.5 has not been edited significanty. The suggestion to include more information about the association between agrichemical exposure and mental health here does not fit in our opinion. This is because this is the essence of our piece and the focus of the results section.

RR2: 
Aim of the study

Aims of the study are not in line with reported results. Aims of the study should be clear and straightforward.

Response:

The authors have taken this under real consideration. As such the results section has been heavily modified. Firstly the aims of the paper have been used to help create the subheadings in the results section i.e. 3.1 is focused on the “Current understandings of agrichemical exposure” whislt 3.2 is focused on the “affect of organophosphate exposure on psychological distress and mental health.” In this way it forces the results section to be aligned with the aims throughout.

RR2: 
Methods

The authors should consider developing the systematic search string and carrying out a systematic review. Grey literature sources might be considered as well.

Methodological check of included papers should be included in methods.

Response:

As described above

RR2: 
Results

Result section is too comprehensive and not completely in line with aims of this study. For instance, Section 3.1. focuses on organophosphate pesticides in poor health outcomes, which includes much more than mental health. The same is valid for some other subsections. Aims of this study focus on mental health outcomes. Therefore, results should be reported in a way that they correspond to the aim of this study.

Response:
The authors have taken this under real consideration. As such the results section has been heavily modified. Firstly the aims of the paper have been used to help create the subheadings in the results section i.e. 3.1 is focused on the “Current understandings of agrichemical exposure” whislt 3.2 is focused on the “affect of organophosphate exposure on psychological distress and mental health.” In this way it forces the results section to be aligned with the aims throughout.

RR2: 
Discussion

Similar to Result section, discussion is very broad and general, drifting away from the aims of this study. In my opinion, it should be revised substantially and focus on the aims of this study, i.e. “The piece herein seeks to 1) 138 investigate the current understanding of agrichemical exposure particularly in the less studied area 139 of farmer mental health, 2) and to gain insight into how chronic (low-level) exposures impinge on 140 psychological distress, to potentially guide screening, prevention and policy developments going 141 forward.”

By carrying out systematic literature review with clear outcome measures, authors will have the opportunity to structure their result and discussion sections in a clear manner, meet scientific standards and deliver quality research with more valid and reliable evidence.

Response:

The discussion section has also been heavily formatted with removal of two paragraphs in its entirety. There is addition of a new paragraph at the end of the section to expand on the discussion about how this research may aid development of public health policies using examples from Australia. The conlusion is edited to be extremely concise and directed towards the aims.

As discussed above, the authors do not feel it necessary to conduct a systematic review. This piece was developed as a scoping review in a broad manner given the lack of research in this field. Itis a topic which draws on a broad range of fields including from the areas of agriculture, mental health, occupational health, and social/behavioural science, and this does not lend itself to a systematic review. Instead, it's important to screen a vast range of literature across these fields to gain an appreciation of the topic. 

Nonetheless, even though this isn’t a systematic review, a PRISMA flow diagram has been produced and inserted as figure 1 in the methods section. Finally, two tables displaying the characteristics of the 33 publications utilised in this study have been constructed and inserted as Table 1 and 2 in the methods.

Round  2

Reviewer 1 Report

I congratulate the Authors on their detailed revision. I believe the paper, in its current form, contributes significantly to the literature on pesticide effects.

Author Response

01.03.2019

Editor

International Journal of Environmental Research and Public Health Editorial Office

Dear Editor,

Re: A Pest to Mental Health? Exploring the Link between Exposure to Agrichemicals in Farmers and Mental Health

In light of your response to our initial revised manuscript, we are delighted that you deem the work worthy for publication. We appreciate your time in reviewing our work. 

Once again thank you for your constructive comments and we trust the revised manuscript meets the requirements of the journal and its readership. 

Yours sincerely,

Dr. Nufail Khan

Reviewer 2 Report

The authors have submitted a revised version of their manuscript, whose aim was to carry out a scoping review addressing an interesting and relevant topic of agrichemical exposure and farmers’ mental health. Although authors put an effort in improving the manuscript, they addressed some of the  comments, leaving out some relevant points to consider. More specifically, the authors focused on their rationale not to conduct systematic review and decided to stay with the scoping review, which is completely acceptable. However, there are still some major shortcomings of their study, which are addressed below point by point, which are similar to the first revision. I recommend authors to seriously consider these points in order to improve their manuscript for publication.

1.

The major point for authors’ consideration is the fact that their manuscript is not in line with the aims of this study. More specifically, the aims are described as follows: “ To 1) investigate the current understanding of agrichemical exposure particularly in the less studied area of farmer mental health, and 2) gain insight into how chronic (low-level) exposures impact on diagnosed mental illness, psychological distress and suicide risk, and how this may guide screening, prevention and policy developments going forward”. However, further sections, i.e. results and discussion address health in general and are not specific to mental health.

2.

Introduction

Please consider consistency in terminology related to mental health. I suggest authors decide between “mental health conditions”, “mental disorders” and “mental illness” and stick to it throughout the manuscript. Additionally, please consider a short section on the relevance of the farming sector for national/international economies.

Line 41 - please define biopsychosocial model for health.

Line 43 – please specify which international data, i.e. are they carried out on a respresentative sample. Since lines 43-45 address mental health, which is the aim of this study, the authors should provide more information on these studies, ie. outcomes (ref. 9, 10 and 11)

Lines 63-67 - similarly to my comments above. It would be interesting to have here, if possible, more information on adverse mental health effects.

Lines 70 - 71 – It is unclear what the authors are trying to say. They could describe these contextual pathways. In addition, it is unclear which dichotomous relationship are they referring to. I suggest to revise for clarity.

Sections 1.3 and 1.4. Title of this section refers to mental health but the content of the section is rather health in general and not mental health. In addition, the content of these and some other sections is somewhat disconnected. Authors should ensure that each individual section in their manuscript has a story, i.e. that they are in line with the title and present meaningful statements that are connected to eachother.

Section 1.5 – Aims are now presented in a more clear manner. Last sentence of this section could be moved to introduction and be a part of a small new section on the relevance of the farming sector for national/international economies.

3.

Methods

Although authors provide some improvements on presenting methodology, there are still some considerable methodological unclarities, such as:

·       Lack of information on the selection process (how many papers have been screened, how many rejected., etc.). More specifically, the Prisma diagram does not provide numbers in each step, only the final number of included papers.

·       The selection criteria (inclusion/exclusion) have not been explained sufficiently.

·       One of the inclusion criteria is “Research reported on pesticide effect on health” which is not in line with the aims of the study. Outcomes included mood disorders, suicidality or neurobehavioral issues, which is again not in line with the aims of this study. Please consider defining aims, outcomes, selection criteria and other relevant parts of Methods in a clear manner.

Table 1 and Table 2 present a nice overview of study characteristics, however they belong to the Result section, along with Prisma diagram.

In addition, Table 1 presents “Characteristics of studies published on pesticide exposure and effects on neurobehavioural or psychiatric disturbances of those in agricultural occupations”.

Aims of the study include mental health, diagnosed mental illness, psychological distress and suicide risk, but not neurobehavioral disorders. Neurobehavioral disorders present another group of health conditions that can, but do not have to include mental health conditions. Hence, authors either adjust aims of the study and the entire manuscript (introduction, results, discussion) accordingly to include studies addressing neurobehavioral disorder, or these should be excluded completely.

Table 2 presents “Characteristics of studies published on effect of low dose pesticide exposure on health outcomes, and the effect of pesticide exposure on suicidality”. Again, similarly as Table 1, this table presents health outcomes, as oppose to specifically mental health outcomes. Aims of this study should be deeply considered when conducting any kind of review, and results should be presented accordingly.

Finally, it would be useful to report on methodology adopted (Tables 1 and 2) and how outcomes were measured.

3.

Results

The results section is still missing a clear structure on what does it present, ie. it is not in line with the aims of this study. The authors could start by describing characteristics of the included studies, for readers to get an overview on findings. Prisma diagram, Table 1 and Table 2 belong to this section, as well. Furthermore, I would consider organizing this section to address aims of the study more specifically and clearly. Some sections are still far away from mental health. For instance, section 3.1 does not address mental health at all, but chemical exposure and their patterns. The authors should consider really focusing on mental health, as stated in the title and aims of their study.

4.

Discussion

The authors have improved their Discussion section. However, what I am missing is discussing their most important findings in a more structured manner, followed by limitations of sources included in the scoping review. The authors could consider reframing their discussion, i.e. to discuss results presented in the context of current literature, policy and practice.

Finally, the authors could use the following publication, as a guidance to produce a structured, clear and methodologically sound scoping review:

Peters, M. D., Godfrey, C. M., Khalil, H., McInerney, P., Parker, D., & Soares, C. B. (2015). Guidance for conducting systematic scoping reviews. International journal of evidence-based healthcare, 13(3), 141-146.

Author Response

Please find attach our response.

Round  3

Reviewer 2 Report

Congratulations to the authors on their significant work and revising the manuscript, which is now, in my opinion, presented in a clear, structure manner, providing new findings and significantly contributing to the field.

I have only few minor comments:

Introduction

Line 32. Reference 4 is from 2009, which is not so recent. I suggest either to find a more recent reference or simply add year of the survey in the text.

Line 28-33. Please decide on terminology. In this section, and later throughout the text for clarity.

Line 36-38 “This burden is exemplified in the fact that 45% of Australians aged 16-85 years will be expected 37 to meet the criteria for diagnosis of a mental health condition at some stage in their life, with 20% of 38 Australians experiencing a mental illness in any one year [6].”

I would suggest to use either one phrase, or rephrase first phrase to “diagnosis of mental illness/disorder” and other to “experiencing a mental health condition”, in order to avoid stigmatizing people with mental health conditions.

Line 45. Please name on how many participants was this large study carried out.

Line 47. Please see previous comment.

Line 48-50. “Recent work of…” - Please add year.  

“Higher suicide risk” - in comparison to?

Line 66. - please add year to “recent”

Line 67. “where authors” instead of “where they”

Line 68. “The analysis”

Methods

Line 167-168. As mentioned in the previous review, when following PRISMA guidelines, Flow Chart Diagram should be part of the Result section. In that case, a short paragraph (few sentences) on included studies should be added to Result section (3.1). In addition, that would ensure that readers can follow further text (i.e. reporting on results and discussion more easily and in a structured manner). However, if authors have sound rationale to leave flow chart as it is and not provide more information to the readers,  I leave that decision to them.

References

This is quite extensive reference list. It would not hurt to shorten it, if possible.

Many Success!

Author Response

Please find attached our PDF response
